# Inferring personal intake recommendations of phosphorous and potassium for end-stage renal failure patients by simulating with Bayesian hierarchical multivariate model

**Jari Turkia[1,2] \***, **Ursula Schwab[3,4]**, **Ville Hautamäki[1]**

**1** School of Computing, University of Eastern Finland, Joensuu, Finland, **2** CGI Suomi Oy, Joensuu, Finland, **3** School of Medicine, Institute of Public Health and Clinical Nutrition, University of Eastern Finland, Kuopio, Finland, **4** Department of Medicine, Endocrinology and Clinical Nutrition, Wellbeing Services County of North Savo, Kuopio University Hospital, Kuopio, Finland

☯ These authors contributed equally to this work.
\* jari.turkia@cgi.com

## Abstract

Most end-stage renal disease (ESRD) patients face a risk of malnutrition, partly due to dietary restrictions on phosphorous and, in some cases, potassium intake. These restrictions aim to regulate plasma phosphate and potassium concentrations and prevent the adverse effects of hyperphosphatemia or hyperkalemia. However, individual responses to nutrition are known to vary, highlighting the need for personalized recommendations rather than relying solely on general guidelines. In this study, our objective was to develop a Bayesian hierarchical multivariate model that estimates the individual effects of nutrients on plasma concentrations and to present a recommendation algorithm that utilizes this model to infer personalized dietary intakes capable of achieving normal ranges for all considered concentrations. Considering the limited research on the reactions of ESRD patients, we collected dietary intake data and corresponding laboratory analyses from a cohort of 37 patients. The collected data were used to estimate the common hierarchical model, from which personalized models of the patients' diets and individual reactions were extracted. The application of our recommendation algorithm revealed substantial variations in phosphorus and potassium intakes recommended for each patient. These personalized recommendations deviate from the general guidelines, suggesting that a notably richer diet may be proposed for certain patients to mitigate the risk of malnutrition. Furthermore, all the participants underwent either hospital, home, or peritoneal dialysis treatments. We explored the impact of treatment type on nutritional reactions by incorporating it as a nested level in the hierarchical model. Remarkably, this incorporation improved the fit of the nutritional effect model by a notable reduction in the normalized root mean square error (NRMSE) from 0.078 to 0.003. These findings highlight the potential for personalized dietary modifications to optimize nutritional status, enhance patient outcomes, and mitigate the risk of malnutrition in the ESRD population.

**Data Availability Statement:** All relevant data are within the paper and its Supporting information files. Additionally, all the code and data for fully reproducing this analysis are publicly available in Github repository: https://github.com/turkiaj/inferring-personal-recommendations-for-renal-patients.

**Funding:** The author(s) received no specific funding for this work.

**Competing interests:** The authors have declared that no competing interests exist.

## Introduction

The majority of end-stage renal disease (ESRD) patients receiving dialysis fail to meet the recommended energy and protein intakes, leading to an elevated risk of malnutrition [1]. Paradoxically, these patients often exceed the recommended levels of phosphorus and saturated fats, associated with an increased risk of cardiovascular diseases [2]. While decreased appetite is a common contributing factor to malnutrition [3], the issue may also stem from the general nutritional guidelines provided to ESRD patients. Currently, patients with end-stage renal disease are provided with general restrictions on dietary phosphorus and, in some cases, potassium intakes to maintain optimal plasma concentrations of potassium, phosphate, and albumin. However, an increasing body of knowledge suggests that individual responses to the same nutrition can vary significantly [4–6]. Given the varying individual responses, it becomes evident that personalized dietary approaches are necessary to achieve target plasma concentration ranges, rather than relying solely on general guidelines.

Personalized diets are also endorsed for renal failure patients. The National Kidney Foundation (NKF) in the United States publishes the Kidney Disease Outcomes Quality Initiative (KDOQI) nutritional guidelines [7], which suggest personalized adjustments for dietary phosphorous and potassium intakes to maintain normal ranges of serum phosphate and serum potassium. Implementing these personalized guidelines is a paradigm shift away from fixed general guidelines [8]. However, there is a pressing need for a systematic method to derive such personalized guidance. To address this, a statistical approach that models a patient's diet and individual reactions can provide valuable support to clinical nutritionists in justifying personalized recommendations.

In statistical methodology, *multivariate methods* encompass techniques that simultaneously model multiple response variables, such as several plasma concentrations associated with a given dietary input. These response variables can exhibit direct relations, seeming unrelatedness, or complete independence [9]. Graphical models, such as directed acyclic graphs (DAGs) or Bayesian networks, can be employed to capture the joint distribution of the response variables by treating the current levels of dietary nutrients and their concentration responses as random variables. These graphical models provide a framework for representing the system and enabling efficient modeling, particularly in large systems. The efficiency is achieved with the Markov boundaries determining the necessary set of predictor variables, allowing sparse and computationally efficient modeling [10]. Turkia et al. [11] proposed a Bayesian network approach for modeling the personal effects of nutrients using mixed-effect parameterization [12]. While sparse graphical model offers computational efficiency, it may potentially overlook valuable information regarding the interconnectedness of concentrations. To address this issue, Bottolo and Banterle et al. [13] have provided an efficient Bayesian implementation of *seemingly unrelated regressions* (SURs) [9] within the context of sparse high-dimensional quantitative trait loci discovery. An efficient implementation of SURs can serve as a useful alternative to Bayesian networks, offering effective modeling of interconnected responses and complementing the benefits of graphical models.

To implement personalized nutritional guidance, we developed personalized graphical models [10] for each of the studied end-stage renal disease patients to simulate their individual reactions to diet modifications. Food intake data and corresponding laboratory analyses were collected from the patients, which were then utilized to construct personalized models of their diets and reactions. Our hypothesis was that the examined concentrations would exhibit a rapid response to variations in nutrient intake, potentially allowing us to statistically estimate the impact of nutrients on concentration levels. While there is limited direct knowledge regarding the relationships between plasma potassium, phosphate, or albumin concentrations,

modeling the system using hierarchical seemingly unrelated regression (SUR) allowed for exploratory analysis of potential cross-correlations among these concentrations. Given the limited size of the analyzed data set and the presence of some missing laboratory analyses, the estimated cross-correlations were used to predict the missing values, ensuring the utilization of all available data. It is worth noting that the patients in our study underwent different treatments; either hospital, home, or peritoneal dialysis. To account for this treatment grouping, we included it as a nested level in our hierarchical model of responses. The final personalized models incorporate both the current estimations of the patients' diets and the hierarchically estimated individual responses, thereby representing each patient's unique composition of plasma concentrations.

In addition to providing insights into the current composition of concentrations, these personalized graphical models can be utilized to infer targeted nutritional guidance. One of our key contributions is the development of an algorithm that generates personalized diet proposals, taking into account specific conditions related to selected plasma concentration limits, and providing individualized intake recommendations. Optimal concentration limits vary based on factors such as age and sex, and our analysis incorporates personalized limits for enhanced accuracy. It is important to note that several diet recommendation options may satisfy the given conditions with equal probability, and the selection of the ideal option requires the expertise of a clinical nutritionist who can consider other personal factors. In our analysis, the objective is to identify the minimum necessary restrictions on phosphorus and potassium intake for patients with renal failure, aiming to provide dietary recommendations that are as accommodating as possible.

Our primary objective in this study is to present a Bayesian approach for estimating hierarchical graphical models of personal nutritional behavior, accompanied by an algorithm to infer personalized diet recommendations. We apply these methods to a cohort of patients with end-stage renal disease, highlighting the methods' effectiveness in tailoring phosphorus and potassium intakes to each patient's specific needs. The results of our study reveal considerable variations in recommended intakes, enabling greater dietary flexibility for certain patients while mitigating the risk of malnutrition. By demonstrating the potential of personalized graphical models and individualized diet recommendations, this work contributes to the advancement of personalized nutrition strategies for patients with renal failure.

## Dialysis patient data

We recruited end-stage renal disease (ESRD) patients for this nutritional study at Kuopio University Hospital dialysis center; 15 women and 22 men participated ($n = 37$). We considered all patients who were in dialysis treatment and healthy enough to participate. The data collection took place from March to September 2018. The ages of the patients ranged from 26 to 81 with an average age of 61. The data consist of food records and laboratory analyses from the same time periods. For each patient, we performed two observations three months apart. On both occasions, the patients were interviewed about their diet in the past 48 hours (48-hour recall method), and the nutrients of reported diets were calculated with Aivodiet software (v. 2.0.2.1, Aivo Finland, Turku). The interview dates were selected so that the patients had their regular laboratory tests at the hospital within a week from the interview, either close before or after the laboratory test. The average actualized difference between interviews and laboratory tests was five days, but this delay was over a week for nine patients, and for five of them, as long as three weeks. The laboratory tests always occurred before the dialysis treatments so that the treatment did not affect the test. Table 1 presents the analyzed nutrients with patients' average levels as well as personal minimums and maximums within the data. Table 2 displays medications and

**Table 1. Nutrient predictors of the model.**

| Nutrient | Study avg. (min-max) |
|---|---|
| Carbohydrates, E% | 43.6 (27.1—63.6) E% |
| Fat E% | 38.9 (23.4—54.1) E% |
| Monounsaturated Fatty Acids, E% | 14.7 (5.6—25.1) E% |
| Polyunsaturated Fatty Acids, E% | 7.1 (2.2—15.8) E% |
| Protein, E% | 15.1 (9.2—22.4) E% |
| Saturated Fatty Acids, E% | 13.7 (5.9—24.5) E% |
| Fiber | 17 (5—42) g/d |
| Protein, g/kg | 0.8 (0.2—2.1) g/kg/d |
| Energy, kcal/kg | 21.8 (5.6—58.6) kcal/kg |
| Calcium | 570 (123—1741) mg/d |
| Sodium | 2588 (813—5487) mg/d |
| **Phosphorous** | **1042 (304—2184) mg/d** |
| **Potassium** | **2785 (1026—5713) mg/d** |
| Salt | 6560 (201—13863) mg/d |
| Water | 1804 (601—3613) ml/d |
| Vitamin D | 8 (0—31) ug/d |

The amount of fatty acids, carbohydrates, and protein is considered as % of total energy intake (E).

other personal details that were used to predict the plasma concentration levels. One notable detail is the type of dialysis treatment, which was hospital hemodialysis ($n = 21$), home hemodialysis ($n = 9$), or peritoneal dialysis ($n = 7$). The dialysis type was also included as a predictor for intake recommendations.

Laboratory tests for renal patients included several measurements, from which concentrations of plasma potassium (P-K), fasting plasma phosphorous (fP-Pi), and plasma albumin (P-Alb) were selected as targets of this analysis for exploring the possibility of less restricted phosphorous and potassium intake. The selected predictors were assumed to reflect the composition of these concentrations; all the energy nutrients, vitamin D, minerals, and fluids. Also, the selected medications were known to directly affect the concentrations. Patients fasted before the laboratory tests although there were analyses that did not require fasting. The

**Table 2. Personal details that are used as predictors.**

| Personal detail | Percentage of patients |
|---|---|
| Act. D-vit | 43% |
| Blood lipid medication | 68% |
| Diabetes medication | 51% |
| Phosphate binder med. | 22% |
| Renavit | 97% |
| Gender | 41% female |
| Home hemodialysis | 24% |
| Hospital hemodialysis | 57% |
| Peritoneal dialysis | 19% |

Of these predictors, the type of dialysis treatment (home hemodialysis, hospital hemodialysis, and peritoneal dialysis) is used to form a nested level of hierarchy in the model to estimate their effects on other nutrition and medication.

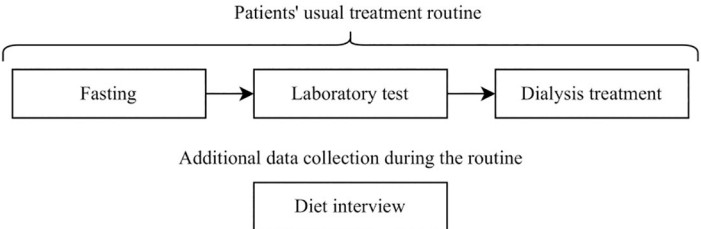

**Fig 1. Data collection during the patients' usual treatment routine.** The figure is drawn with diagrams.net (v 22.0.6, https://app.diagrams.net).

schema for data collection is outlined in Fig 1. Patients' targeted normal ranges are reported in Table 3. These ranges adhere to the statements of the hospital laboratory staff who analyzed the concentrations. The numbers provided a normal range for fasting plasma phosphate (fP-Pi) only for healthy persons, but all end-stage renal disease patients (CKD 5D) in this analysis were undergoing dialysis, and for them, a normal range of 1.13–1.78 mmol/l is the target, according to KDOQI guidelines [14]. The laboratory staff stated that the normal range of plasma albumin (P-Alb) varies depending on the patient's age, and this varying target was considered in our recommendation. S1 Fig presents potassium and phosphorous intake levels from the collected food records and their observed effects on these concentrations; the recommended normal ranges are denoted with white areas. General intake recommendations [8] for potassium (2500 mg/d) and phosphorous (1000 mg/d) are indicated with black vertical lines. The 2020 update of the KDOQI Clinical Practice Guidelines [7] omitted these strict general guides and advised adjusting dietary phosphorus and potassium intakes to maintain normal ranges of serum phosphate and potassium concentrations.

A research permit for this study was granted by Kuopio University Hospital Research Assistance Center ("KYS Tiedepalvelukeskus" in Finnish) which is an Institutional Review Board of Kuopio University Hospital. The permit waived ethical approval because participants were not subjected to any additional procedures, visits, or tests beyond their usual routine. Instead, only authorized researchers were allowed to handle sensitive and identifying information according to the permit. Patients were given information about the study in writing and orally before starting the study. Before the interviews and data collection, informed written consent was obtained from all participants. The patients had the right to stop their participation in the study at any stage. Participation in the study did not harm the patients and did not affect their treatment. Interviews with the patients took place during their dialysis treatment, and the

**Table 3. Personalized normal ranges for plasma concentrations depending on the age of the patient.**

| Plasma concentration | Normal range | Target group |
| --- | ---: | ---: |
| Plasma potassium (P-K) | 3.4–4.7 mmol/l | Everyone |
| Fasting plasma phosphate (fP-Pi) | 1.13–1.78 mmol/l | Dialysis patients, CDK 5D |
| Plasma albumin (P-Alb) | 36–48 g/l | 39 years or younger |
| Plasma albumin (P-Alb) | 36–45 g/l | 40–69 years old |
| Plasma albumin (P-Alb) | 34–45 g/l | 70 years and older |

Normal plasma potassium and albumin ranges are used according to the hospital laboratory that analyzed the concentrations. All end-stage renal patients in this analysis are in dialysis (CKD 5D) and for they a normal range 1.13–1.78 mmol/l is targeted according to National Kidney Dialysis Outcomes Quality Initiative (KDOQI) guideline. [14].

participants granted permission to utilize their laboratory analyses for the study. Research data were treated confidentially; no individual patient could be identified from the collected data or the results of this study, and thus, no dietary modifications have been done based on this study. The authors of this study cannot identify the participating patients from the collected data or by any other means. Patient IDs in the article figures and tables are only used for patient reference in the results and they cannot be used to identify the study subjects.

## Inferring personalized recommendations for nutrient intake

In this section, we present a method used for determining personalized recommendations for nutrient intake. We especially focused on a case where some of the nutrients in a patient's diet were modified while the remaining nutrients maintained their current intake. These recommendations were designed to ensure that the patient's plasma concentrations fall within predefined normal ranges with a desired level of confidence. Achieving this desired confidence level relies on accurate estimates of the patient's current nutrient intake and the effects of nutrients. It is crucial to have a high level of confidence in these estimates before relying on the recommendations. To infer personalized nutrient intake recommendations, we began by constructing personal graphical models that replicated the observed concentrations based on the patient's current dietary intake. These personal models were then utilized in simulating the optimal intake adjustments predicted to maintain normal concentration ranges. A summary of variables, indices, and other notations of the method description is given in S4 Table.

### Personalized graphical models for plasma concentrations

In our analysis, we constructed a directed graphical model [10]$G_k$ for each patient $k$. These graphical models represented joint conditional distributions over random variables for concentration levels $Y_{km}$, levels of nutrients in diet $X_{kj}$, and effects of those nutrients in multiple levels of detail. The graph is illustrated in Fig 2A, including the effects of nutrients in the general ($\beta_{kjm}$), dialysis treatment ($g_{ljm}$), and personal levels ($b_{kjm}$). Every concentration $m = 1, \ldots,$ $M$ was conditioned with the same $j = 1, \ldots, p$ nutrients and personal treatment type $l = 1, \ldots,$ $L$; hence the edges of the graph are directed towards the concentration levels $Y_{km}$.

The concentration random variables $Y_{kmi}$ were assumed to follow gamma distribution that allows only positive values and skews to the right, thus allowing occasional values that were considerably above average [15] as follows

$$Y_{kmi}|\alpha_m, \mu_{kmi} \sim \text{Gamma}\left(\alpha_m, \frac{\alpha_m}{\mu_{kmi}}\right), \quad k = 1, \cdots, K, \ m = 1, \cdots, M, \ i = 1, \cdots, n \quad (1)$$

where $\alpha_m$ was a shape parameter of gamma distribution for concentration $m$. We used an inverse scale parameterization of gamma distribution where its rate parameter was obtained by dividing this shape parameter with an expected value $\mu_{kmi}$ for the $i$th observation of patient $k$. By this formulation, the expected concentration was clearly defined as a linear combination of the amounts of nutrients and other predictors and their effects with

$$\mu_{kmi} = X_{kji}\beta_{jm} + Z_{kji}g_{ljm} + Z_{kji}b_{kjm} \quad (2)$$

that summed the general effect of a nutrient $j$ with the effect variation caused by the dialysis treatment $l$, and finally, the personal variation within the treatment for patient $k$. This formed a nested two-level hierarchical model for repeated personal observations and the effect of treatment that every patient undergoes; either hospital, home, or peritoneal dialysis. Random variable $X_{kji}$ denotes $i$th observation of predictor $j$ for patient $k$, and random variables $Z_{kji}$ denote predictors whose effects were assumed to vary between treatments or patients. In this analysis,

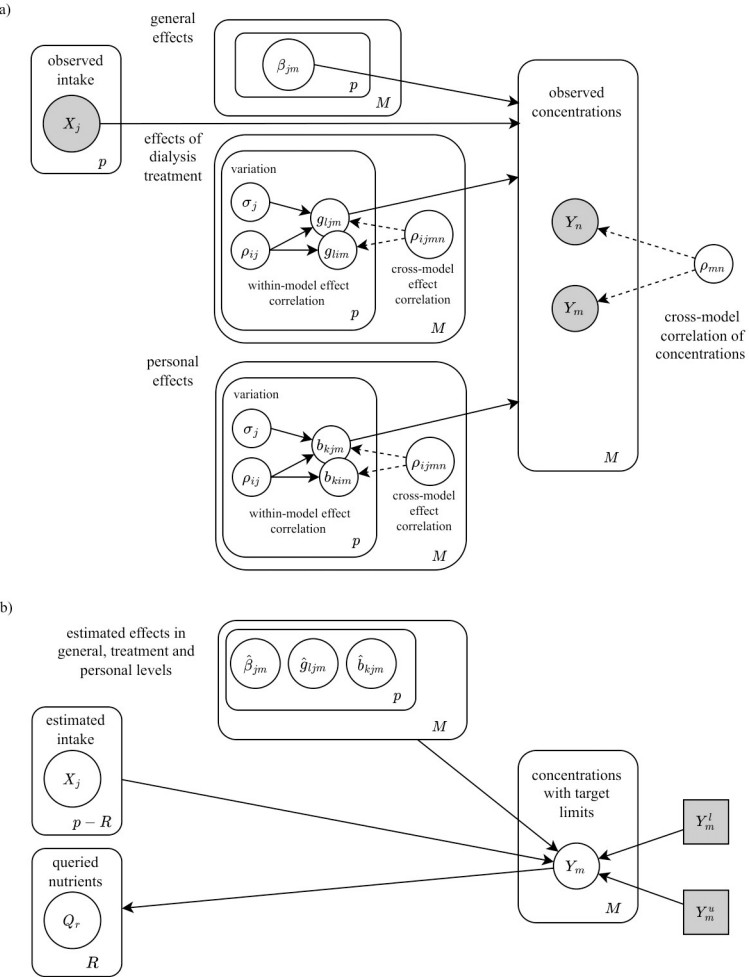

**Fig 2. Structure of the graphical model with general, treatment, and personal level nutrient effects.** A) The hierarchical graphical model for effects of nutrients in general, treatment, and personal levels. B) Information flow of the queried nutrients when conditioned with the current diet, personal effects, and concentration limits. The figure is drawn with diagrams.net (v 22.0.6, https://app.diagrams.net).

all predictors were assumed to vary, and thus $X_{kji}$ and $Z_{kji}$ referred to the same variables, but different sets of predictors could be selected here.

Both parameters of gamma distribution are required to be positive, but we needed to keep the estimated effects of nutrients in an additive scale for direct interpretation. The additive scale can cause problems as it allows also negative values. This was solved by transforming the expected value $\mu_{kmi}$ linearly by adding both sides a constant $c$ which was sufficiently large to move the estimation to strictly non-negative values while not affecting the scale of the estimated effects. For parameter $\alpha_m$, we could apply an exponent link function that enforced non-negative values. It transformed the values to a logarithmic scale, but their direct interpretation was not important for this analysis. The result was a linearly shifted distribution of form

$$Y_{kmi} + c | \exp\alpha_m, \mu_{kmi} + c \sim \mathrm{Gamma}\left( \exp\alpha_m, \frac{\exp\alpha_m}{\mu_{kmi} + c} \right) \qquad (3)$$

where the original distribution $Y_{kmi}$ in Eq (1) was obtained with reverse transformations.

## Estimating nutritional effects with hierarchical seemingly unrelated regressions

The effects of nutrients were estimated by considering the $M$ different concentration distributions in Eq (1) as seemingly unrelated regressions (SUR) [9]. In SURs, there are no direct relationships between the regression models, but their hierarchical effects can be dependent on each other. This allowed us to consider all the concentrations simultaneously even though they are not directly related. The estimation was conducted by formulating all expected value regressions from Eq (2) as one univariate matrix:

$$
\begin{aligned}
\boldsymbol{\mu} &= \mathbf{X}\boldsymbol{\beta} + \mathbf{Z}^{(g)}\mathbf{g} + \mathbf{Z}^{(b)}\mathbf{b}, \\
\mathbf{X} &= \mathbf{I}_{M \times M} \otimes \mathbf{X}_{p \times n}, \\
\mathbf{Z}^{(g)} &= \mathbf{I}_{M \times M} \otimes \mathrm{diag}(\mathbf{Z}_1^{(g)}, \cdots, \mathbf{Z}_L^{(g)}), \\
\mathbf{Z}^{(b)} &= \mathbf{I}_{M \times M} \otimes \mathrm{diag}(\mathbf{Z}_1^{(b)}, \cdots, \mathbf{Z}_K^{(b)})
\end{aligned}
\tag{4}
$$

where the model matrix $\mathbf{X}$ comprises all the predictor variables and was created using a Kronecker product, which is a matrix outer product resulting in a block matrix structure. This pattern follows a block-diagonal arrangement, where the same set of predictors is replicated across all $M$ concentration models. Similarly, block matrices $\mathbf{Z}^{(g)}$ and $\mathbf{Z}^{(b)}$ were formed with Kronecker products to repeat, in a block-diagonal pattern, the predictors for treatment and personal effects for each $1, \ldots, L$ treatment and $1, \ldots, K$ person. These matrices contain the same data but were organized differently for selecting the relevant input for each patient and their respective treatment when multiplied by the effect vectors $\boldsymbol{\beta}$, $\mathbf{g}$, and $\mathbf{b}$.

The treatment and personal effect vectors were drawn from multivariate Gaussian distributions, $\mathbf{g} \sim \mathcal{N}(0, \boldsymbol{\Sigma}_g)$ and $\mathbf{b} \sim \mathcal{N}(0, \boldsymbol{\Sigma}_b)$, that were centered to general effects $\boldsymbol{\beta}$ and had variance-covariance matrices $\Sigma_g$ and $\Sigma_b$. The variance-covariance matrices were defined with diagonal matrices $\mathbf{T}_g = \mathrm{diag}(\sigma_{g_1}, \ldots, \sigma_{g_l})$ and $\mathbf{T}_b = \mathrm{diag}(\sigma_{b_1}, \ldots, \sigma_{b_k})$ containing standard deviations for the effects as well as correlation matrices $\mathbf{C}_g$ and $\mathbf{C}_b$ that decompose into triangular Cholesky decomposition matrices $\mathbf{L}_g$ and $\mathbf{L}_b$ as follows

$$
\begin{aligned}
\boldsymbol{\Sigma}_g &= \mathbf{T}_g \mathbf{C}_g \mathbf{T}_g' = \mathbf{T}_g \mathbf{L}_g \mathbf{L}_g' \mathbf{T}_g', \\
\boldsymbol{\Sigma}_b &= \mathbf{T}_b \mathbf{C}_b \mathbf{T}_b' = \mathbf{T}_b \mathbf{L}_b \mathbf{L}_b' \mathbf{T}_b'.
\end{aligned}
\tag{5}
$$

This model formulation produces the correlation matrices $\mathbf{C}_g$ and $\mathbf{C}_b$ whose structures contain similar the within-model correlations at diagonal blocks and cross-model correlations at off-diagonal blocks:

$$
\mathbf{C} = \begin{bmatrix}
\mathbf{D}^{(1)} & \mathbf{C}^{(12)} & \cdots & \mathbf{C}^{(1M)} \\
\mathbf{C}^{(12)'} & \mathbf{D}^{(2)} & \cdots & \mathbf{C}^{(2M)} \\
\vdots & \vdots & \ddots & \vdots \\
\mathbf{C}^{(1M)'} & \mathbf{C}^{(2M)'} & \cdots & \mathbf{D}^{(M)}
\end{bmatrix}
\tag{6}
$$

where matrix blocks $\mathbf{D}^{(m)}$, $m = 1, \ldots, M$, denote correlations of effects within a concentration $m$ and blocks $\mathbf{C}^{(nm)}$ denote effect correlations between concentrations $m$ and $n = 1, \ldots, M$. The correlation matrix blocks are transposed across the diagonal. This system is illustrated as a

personal graphical model in Fig 2A, which depicts these within-model and cross-model correlations with random variables $\rho_{ijmn}$ of the graphical model.

The estimated variance-covariance matrices provided all the necessary information for predicting the personal effects also for new patients with previously unseen observations. This approach was employed in the cross-validation predictions of this study. However, in general, it's important to exercise caution with out-of-sample predictions. The predictions were generated using Eq (4) by placing the new observations in data matrices. Personal effect predictions $\hat{\mathbf{b}}$ follow distribution $\hat{\mathbf{T}}_b\hat{\mathbf{L}}_b\mathbf{z}$ where $\mathbf{z}$ is a standard normal distribution $\mathcal{N}(\mathbf{0}, \mathbf{I})$ that is scaled with estimated Cholesky decomposition $\hat{\mathbf{L}}_b$ and the standard deviation of the effects $\hat{\mathbf{T}}_b$. In the cross-validation, we utilized the expected values of this predictive distribution as they are the most probable effects to produce the observed concentrations with the given predictors.

Finally, we acknowledged that some nutrients were likely to have collinearly similar effects on concentrations, which could result in inaccurate parameter estimations [16]. To decorrelate the independent variables, a QR decomposition [17] was used to deconstruct the model matrix of common effects $\mathbf{X}$ into an orthogonal matrix $\mathbf{Q}$ and an upper-triangular matrix $\mathbf{R}$. This deconstruction was further developed into a thin QR decomposition, $\mathbf{X} = \mathbf{Q}^*\mathbf{R}^*$, which is equivalent but more computationally efficient, as $\mathbf{Q}^* = \mathbf{Q}\sqrt{n-1}$ and $\mathbf{R}^* = \frac{1}{\sqrt{n-1}}\mathbf{R}$ were scaled down by the number of observations $n$. By estimating parameters $\boldsymbol{\beta}^{*(m)} = \mathbf{R}^*\beta^{(m)}$, significantly less correlated common effect coefficients $\boldsymbol{\beta}^{(m)} = \mathbf{R}^{*-1}\beta^{*(m)}$ needed to be calculated as the matrix $\mathbf{Q}$ is orthogonal with independent columns.

## Probabilistic inference with personal generative models

Personalized models were made generative by defining proper prior distributions for all parameters, including the nutrient intakes in each patient's diet. Scholars recognize [18] that intake levels follow positive and right-skewed distributions, such as log − normal or gamma distributions. However, intake distribution is different than estimating the current level of intake as required in this task. Normal distribution is a better option here, as it captures the variance of the dietary intake observations but reduces the unseen intake levels. We assumed that the mean of observations was the expected value of the intake, and the standard deviation of observations provides uncertainty:

$$X_{kj} \sim \mathcal{N}\left(\hat{\mu}_{kji}, \hat{\sigma}^2{}_{kji}\right) = \mathcal{N}\left(\overline{X}_{kji}, S^2_{kji}\right),\ k = 1, \cdots, K,\ j = 1, \cdots, J,\ i = 1, \cdots, n \qquad (7)$$

We denote the resulting, fully estimated personalized graph with $\hat{G}_k$. Personal estimations of the nutritional effects were obtained at the personal, most detailed, level of the hierarchical model, allowing for fine-grained adjustments that mitigate possible bias inherent in upper levels of the model. In this graph, the concentration estimates $\hat{Y}_{km}$ were assumed to settle near their observed values when using the current nutrient intakes and the estimated effects of nutrients. This allowed analysis of the personal compositions of the concentrations, and in addition, we could also condition the graph for any of its variables. This *probabilistic inference* [19] for intake recommendations is illustrated in Fig 2B where the edges connecting to the queried nutrients were now reversed toward them. We denote this subset of $r = 1, \ldots, R$ nutrients that were queried for recommendations with $Q_{kr}$ and the rest of $j = 1, \ldots, J − R$ nutrients continue to be denoted with $X_{kj}$. They maintained their current personalized estimations while the conditioned nutrient variables were defined with new proposal distributions that allowed all possible healthy levels for these nutrients. In this work, we used uniform distributions with

nutrient-specific limits, but a more informative distribution could also be used. By denoting this personalized graph with modified variables as $\hat{G}_k^*$, the conditional probability for the intake recommendation can be formulated as

$$p(Q_{kr}|\hat{G}_k^*, \, P(Y_m^l < Y_m < Y_m^u) > c), \, r = 1, \cdots, R, \, m = 1, \cdots, M \qquad (8)$$

where the queried nutrients $Q_{kr}$ are conditioned with values that cause all plasma concentrations $Y_m$ to reside within their predefined lower and upper bounds $Y_m^l$ and $Y_m^u$ with probability $c$.

## Algorithm for recommendation sampling

Algorithm (1) estimates the conditional recommendation distribution in Eq (8) by acceptance sampling. In summary, the algorithm samples diet proposals from the previously defined variables $Q_{kr}$ and $X_{kj}$, and evaluates the probability of how confidently this diet proposal produces plasma concentrations that reside within the requested limits. The distribution of these levels of confidence forms the personal recommendation for the queried nutrients $Q_{kr}$. For clear reporting and systematic comparison between patients, the algorithm returns 2.5%- and 97.5%-quantile values for every recommended nutrient; the example of the full distribution of Eq (8) can be seen in Fig 3.

**Algorithm 1**: Multivariate Acceptance Sampling for Intake Recommendation

```
Input:
Ĝ*,        ▷ Conditioned graphical model including variables Qᵣ with
priors
Yₘˡ,Yₘᵘ,     ▷ Targeted lower and upper limits for concentrations Yₘ
lₓ,        ▷ Quantile of intake random variables (X̂ⱼ in Ĝ*) used in
estimation
lᵦ,        ▷ Quantile of nutrient effects (β̂ⱼₘ in Ĝ*) used in estimation
S,         ▷ Number of drawn samples from the queried nutrients
c          ▷ Targeted confidence level
Output:
P(q̃ᵣₛ)      ▷ Probabilities for queried nutrients reaching the concen-
tration targets
Qᵣᵐⁱⁿ,Qᵣᵐᵃˣ,    ▷ 95%-quantile of R-dimensional recommendation
distribution
μ̂ₘ�q⁰ ▷ Expected concentration m without the intake of queried nutrients
Pₘᵐᵃˣ       ▷ Maximum probability for reaching target for concentration m
Pᵐᵃˣ        ▷ Maximum probability for reaching all the concentration
targets
Algorithm RecommendationSampling (Ĝ*, Yₘˡ, Yₘᵘ, lₓ, lᵦ, S, c):
Qᵣᵐⁱⁿ,Qᵣᵐᵃˣ,μ̂ₘq⁰,Pₘᵐᵃˣ,Pᵐᵃˣ
1  Pᵐᵃˣ ← 0;
2  Pₘᵐᵃˣ ← 0, m = 1,...,M;
3  p ← number of nutrient random variables X̂ⱼ in Ĝ*
4  x̂ⱼ ← lₓ-quantile value of X̂ⱼ in Ĝ* for all j = 1, ..., p − R;
5  β̂ⱼₘ ← lᵦ-quantile value of β̂ⱼₘ in Ĝ* for all j = 1, ..., p, m = 1, ..., M;
6  α̂ₘ ← α-parameter of concentration distribution m in Ĝ*;
7  foreach concentration m in Ĝ* do
8      μ̂ₘq⁰ ← x̂ⱼβ̂ⱼₘ, j = 1,...,p − R;
9      lₘ ← min(μ̂ₘq⁰, Yₘˡ);
10     uₘ ← min(μ̂ₘq⁰, Yₘᵘ);
11 end
12 for samples s = 1, ..., S do
13     q̃ᵣₛ ← DrawSample(Qᵣ = q̃ᵣ, lₘ < μₘq⁰ + q̃ᵣₛ < uₘ, m = 1,...,M), r = 1,...,R;
```

```
14   foreach concentration m in Ĝ* do
15       μ̂_ms ← μ̂_m^q0 + q̃_rs β̂_im^p, r = 1,...,R, i = p − R,...,R;
16       Y_ms ← Gamma(α̂_m, α̂_m/μ̂_ms);
17       P_ms ← P(Y_m^l ≤ Y_ms ≤ Y_m^u);
18       P_m^max ← max(P_m^max, P_ms);
19   end
20   P(q̃_rs) ← min(P_ms, m = 1,...,M);
21   P^max ← max(P^max, P(q̃_rs));
22 end
23 for queried nutrients r = 1, ..., R do
24   C_rs ← q̃_rs, P(q̃_rs) > c, s = 1,...,S;
25   Q_r^min ← q̃, P(C_rs = q̃) ≤ 2,5%, s = 1,...,S;
26   Q_r^max ← q̃, P(C_rs = q̃) ≥ 97,5%, s = 1,...,S;
27 end
```

The steps of the Algorithm (1) are explained in detail. To provide transparent reasoning, the algorithm tracks the overall and concentration-specific probabilities of reaching the targets, and these are initialized to 0 in Steps (1) and (2). The algorithm iterates extensively on the nutrient random variables and their effects, and their number in the graph is counted in Step (3). To manage the uncertainty in the potentially wide distributions of these variables, the algorithm uses point estimates in their quantile values. By default, the expected values, in the mean of the distributions, are used, but tail quantiles are also useful in sensitivity analysis. The used quantiles are provided by parameters $l_x$ and $l_\beta$ and they are used to acquire point estimates of these random variables in Steps (4) and (5).

To optimize the sampling, the loop in Step (6) calculates concentration-specific parameters that remain fixed during the sampling. Step (7) calculates the expected values of concentrations without the effect of the queried nutrients. This provides constant baselines of concentration levels $\hat{\mu}_m^{q0}$ that do not change during the sampling. These constants are used in Steps (8) and (9) to determine the well-defined minimum and maximum limits for the sampled system. The expected values $\hat{\mu}_m^{q0}$ are used instead of the required concentration limits if they are lower than the concentration's lower limit or higher than the concentration's upper limit to prevent the sampling from failing.

The main loop in Step (11) draws and evaluates the $S$ number of samples from the queried nutrients. Vectors of $R$ coincidental samples are drawn from $Q_r$ in Step (12); a sampling function that is provided with the queried random variables as well as the lower and upper sampling limits from Steps (8) and (9). The queried nutrients $Q_r$ are assumed to follow suitable proposal distributions for sampling. For actual sampling, we used No-U-Turn sampling (NUTS) [20] algorithm to draw the samples from this constrained posterior. This algorithm is efficient in sampling these multidimensional distributions.

After a diet proposal is sampled, the loop in Step (13) iterates all the concentrations and calculates their expected values in Step (14) as well as the full posterior distribution in Step (15). The posterior distribution is used in Step (16) to evaluate the confidence level for each diet proposal by assessing the amount of cumulative density between the concentration limits. The maximum of the reached confidence level is stored for each concentration in Step (17) for diagnostic purposes.

The algorithm seeks intake proposals that are estimated to take all the concentrations within the normal ranges, between $Y_m^l$ and $Y_m^u$, with the minimum confidence level $c$; therefore, a minimum of achieved concentrations' confidence level is stored for the current proposal in Step (19). Step (20) stores also the maximum overall confidence level for diagnostic purposes.

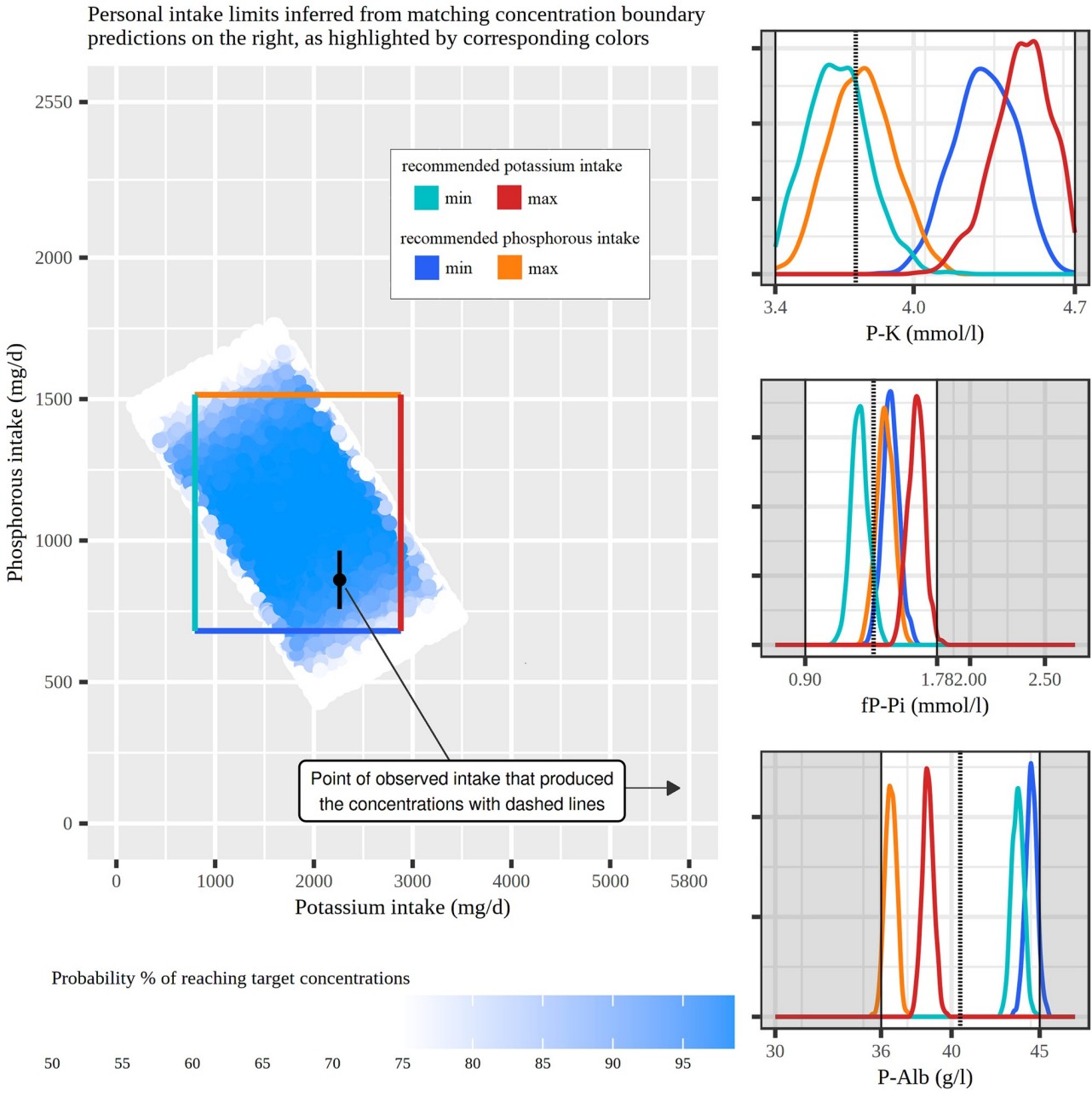

**Fig 3. Detailed intake recommendation for Patient 36.** The intake recommendation for Patient 36 on the left shows the inferred diet configurations predicted to produce normal ranges of potassium (P-K), phosphate (fP-Pi), and albumin (P-Alb) concentrations. These limits are marked with white areas in the concentration panels on the right. The black point in the middle of the intake plot represents the patient's current potassium and phosphorous intake producing the current concentrations shown with dashed vertical lines. The colored rectangle shows the limits of intake constrained by the correspondingly-colored concentration predictions at the boundaries of the normal ranges. The figure is plotted with ggplot2 package for R language (v 3.4.1, https://ggplot2.tidyverse.org).

Finally, in Steps (22)-(25), minimum and maximum recommendations are picked for all the queried nutrients. In this implementation, we report 2.5% and 97.5% quantile values of a subset $C_r$ for the queried distributions that have confidence levels in sampled proposals $P(\tilde{q}_{rs}) > c$ for all the concentrations. These quantile values are returned as results in $Q_r^{min}$ and $Q_r^{max}$.

## Implementation

The probabilistic models for both hierarchical effects of nutrients and personalized recommendations were implemented with the probabilistic programming language *Stan* [21] and the estimation of the models was completed through Stan's implementation of No-U-Turn sampling [20]. Personalized graphical models that use the effect estimations from the hierarchical model were constructed with a custom R-code and iGraph R-package [22]. These graphs represent random variables as nodes and their connections as edges. The estimated distributions of random variables are stored in the properties of the nodes, which allowed us to fully execute Bayesian inference over the graph. The inference for personalized recommendations through Algorithm (1) was implemented in a custom R-code that filtered the samples that were drawn from the recommendation model.

## Data application

We applied Algorithm (1) to infer personalized recommendations of phosphorous and potassium intakes for end-stage renal disease patients based on the collected data. Our main results in Fig 4 demonstrate considerable differences between patients in the recommended intakes of phosphorous and potassium. Some patients could exceed the general recommendations while others should limit intakes of either or both nutrients to levels below the general recommendations. For many patients, though, the required confidence level of personalized recommendations could not be achieved by modifying potassium or phosphorous alone. In particular, the required lower limit of plasma albumin was unreachable for many patients. Detailed results and simulations with personalized models are elaborated next.

## Creating personalized models for all patients

Our main interest was to personalize patients' diets to reach the suggested normal ranges of plasma potassium (P-K), fasting plasma phosphorous (fP-Pi), and plasma albumin (P-Alb) in Table 3. However, seven of 37 patients had missing plasma albumin measurements. To overcome the missing data, we created personalized models for these patients to predict the missing values before the main analysis. The hierarchical model used in Eq (1) learns typical reactions from patients who have all three concentration measurements available and then generalizes the reactions for patients who have missing measurement values. In the final data set, we imputed the predicted albumin concentrations for these patients. We assumed that this data imputation provided a minimal bias in data and allowed us to use all of the available measurements.

The main analysis was conducted by first estimating the system of hierarchical nutrient effect models in Eq (1) with the three concentration targets and then constructing the personalized graphical models $\hat{G}_k$ for $k = 1, \ldots, 37$ patients. In these personalized models, patients' current nutrient intake variables $X_{kj}$ were estimated by using Eq (7). The collected intake data are described in Table 1. The table shows also the differing intakes of these nutrients within the study. The input data also included personal factors, such as medication, which are described in Table 2. The estimated effects of phosphorous and potassium are provided in Table 4 at all hierarchically modeled levels. The effects of all considered nutrients are presented in S3 Table, which is arranged in decreasing order of between-treatment variations; with this arrangement, the effect of water on plasma albumin is noticeably different between treatments. Table 5 highlights the strongest effects with high individual variation. The table shows also the estimated variations of the effects between dialysis treatments ($\hat{\sigma}_g$). It is notable that most varying effects are related to plasma albumin concentration. Further reasoning for the effect predictions is available in S2 and S3 Figs. The correlation plots of dialysis treatments and

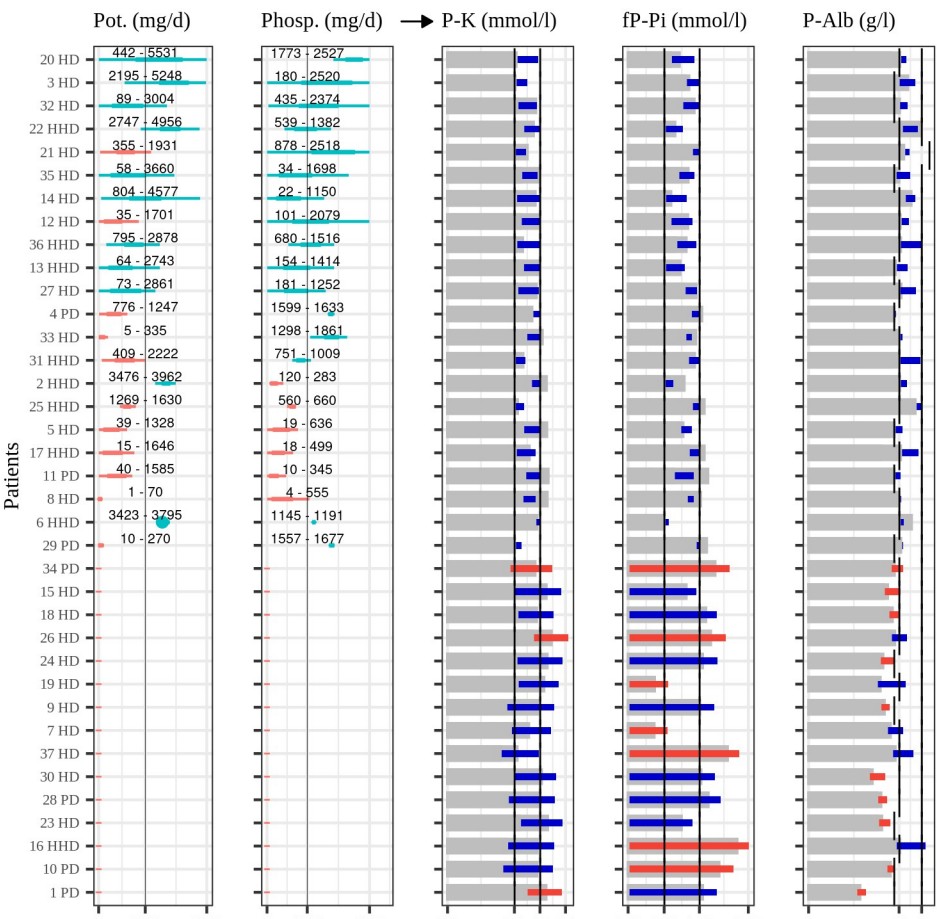

**Fig 4. Personal algorithmic recommendations of potassium and phosphorous intake and the corresponding concentration predictions.** The figure illustrates, in the two leftmost panels, personalized recommendations for potassium and phosphorous intake derived from Algorithm (1), along with the matching concentration predictions in the rightmost panels. Each row corresponds to a patient with a numeric label and the type of dialysis treatment they are undergoing (HD = hospital hemodialysis, HHD = home hemodialysis, PD = peritoneal dialysis). The colored bars within the intake recommendation utilize blue and red colors to denote whether or not the recommendations surpass the general guidelines of 2500 mg/d for potassium and 1000 mg/d for phosphorous, thus allowing a personally richer intake. Above these bars, numeric values represent the 95% credible interval for each recommendation. These intake recommendations are predicted to produce the concentrations depicted in the right panels. Here, the grey bars represent the estimated concentrations disregarding the impacts of potassium and phosphorous intake ($\mu_{q0}$). When Algorithm (1) provides a recommendation with over 90% confidence ($P^{max} > 90\%$), the predicted concentrations within personal limits are highlighted in dark blue. In cases where the recommendation lacks sufficient confidence, the recommendation is excluded, and the entire achievable concentration range is shown with red and blue bars. The red color indicates the concentration whose limits cannot be attained, and for many patients, it is the lower plasma albumin limit. This figure was generated using the ggplot2 package for the R language (v 3.4.1, https://ggplot2. tidyverse.org).

personalized matrices in these figures form a network of associated effects for the sample population. S1 and S2 Tables provide detailed figures of effects that have the highest correlations in levels of treatment and individual patients.

## Comparing simulated personalized recommendations

Personalized recommendations were acquired by executing Algorithm (1) for all patients $k = 1, \ldots, 37$, and providing a conditioned personalized graphical model $\hat{G}_k^*$ as an input. The

**Table 4. Effects of potassium and phosphorous between different dialysis treatments and between patients within the same treatment.**

| Nutrient | Conc. | General effect | Home hemodialysis | | | Hospital hemodialysis | | | Peritoneal dialysis | | |
|---|---|---|---|---|---|---|---|---|---|---|---|
| | | | avg | min | max | avg | min | max | avg | min | max |
| Phosphorous | P-Alb | -0.19 [−7.16;6.56] | **-2.10** [−10.73;5.28] | -3.09 [−12.68;4.94] | -1.87 [−12.44;7.81] | **0.27** [−5.57;5.95] | -0.41 [−6.77;6.08] | 1.73 [−4.98;8.63] | **0.80** [−7.31;8.48] | 0.10 [−8.68;8.11] | 1.22 [−6.42;8.81] |
| Phosphorous | P-K | **0.39** [−2.48;2.31] | 0.25 [−1.41;2.02] | 0.15 [−1.57;1.79] | 0.36 [−1.54;2.84] | **0.21** [−1.20;1.69] | 0.10 [−1.37;1.54] | 0.37 [−1.30;2.61] | **0.51** [−1.42;2.55] | 0.45 [−1.44;2.39] | 0.59 [−1.49;2.61] |
| Phosphorous | fP-Pi | 0.08 [−0.90;1.41] | **0.15** [−1.00;1.13] | 0.10 [−1.19;1.23] | 0.21 [−1.14;1.21] | **-0.02** [−0.79;0.79] | -0.09 [−0.95;0.81] | 0.08 [−0.63;0.76] | **0.16** [−0.91;1.43] | 0.05 [−1.22;1.43] | 0.20 [−0.90;1.45] |
| Potassium | P-Alb | -3.14 [−15.40;4.12] | **-3.89** [−9.99;1.30] | -5.04 [−12.20;0.79] | -3.63 [−9.67;2.06] | **-1.06** [−4.53;1.96] | -1.97 [−8.76;2.01] | 0.34 [−3.64;3.96] | **-0.24** [−5.53;5.54] | -1.02 [−7.03;5.17] | 0.27 [−5.92;8.05] |
| Potassium | P-K | -0.46 [−3.35;2.02] | **-0.06** [−1.48;1.55] | -0.32 [−2.14;1.32] | 0.14 [−1.57;1.76] | **0.14** [−0.58;1.02] | -0.11 [−1.40;1.27] | 0.50 [−0.52;1.64] | **-0.21** [−1.66;0.98] | -0.51 [−2.22;1.17] | 0.16 [−1.50;2.45] |
| Potassium | fP-Pi | 0.07 [−0.88;0.88] | **0.00** [−0.72;0.76] | -0.05 [−0.82;0.77] | 0.04 [−0.72;0.78] | **0.06** [−0.34;0.52] | 0.03 [−0.51;0.53] | 0.09 [−0.35;0.56] | **0.17** [−0.56;0.97] | 0.15 [−0.68;1.03] | 0.21 [−0.48;1.05] |

The table shows the effects of potassium and phosphorous on concentrations ($m = 1, \ldots, 3$) for analyzed patients ($k = 1, \ldots, 37$) in all three additive levels of the model. General effects ($\hat{\beta}_{jm}$) show the sample mean of the effect that is shown to vary between patients in home hemodialysis, hospital hemodialysis, and peritoneal dialysis. The first column of each dialysis type (avg) shows the typical effect of the treatment ($\hat{\beta}_{jm} + \hat{g}_{ljm}, l = 1, \ldots, 3$) that can further vary personally. The minimum and maximum of these personal effects ($\hat{\beta}_{jm} + \hat{g}_{ljm} + \hat{b}_{kjm}$) are shown within each treatment. For every effect, 90%-credible intervals are shown under the expected value at the posterior mean.

**Table 5. Strongest and personally most varying effects of nutrients and medication.**

| Effect | Expected effect strength | | | |
|---|---|---|---|---|
| | General effect | Min. personal | Max. personal | Treatment var. |
| Renavit → P-Alb | 10.04 [-12.1; 24.48] | 7.28 [-12.78; 21.69] | 12.77 [-7.93; 28.05] | 4.66 [0.39; 12.44] |
| Potassium → P-Alb | -3.14 [-15.4; 4.12] | -5.04 [-12.2; 0.79] | 0.34 [-3.64; 3.96] | 4.85 [0.32; 12.99] |
| Salt → P-Alb | 1.60 [-8.59; 10.81] | -0.81 [-12.25; 8.04] | 4.52 [-6.23; 12.73] | 6.19 [1.1; 17.17] |
| Phosphate binder med. → P-Alb | -2.78 [-9.87; 4.87] | -4.26 [-18.77; 8.61] | 0.84 [-10.26; 12.01] | 5.74 [0.74; 16.86] |
| Fat, E% → P-Alb | 3.19 [-14.81; 19.59] | 1.52 [-17.42; 20.12] | 4.14 [-12.29; 19.78] | 4.42 [0.77; 10.57] |
| Gender → P-Alb | 1.67 [-5.48; 10.14] | -0.53 [-11.15; 7.42] | 4.13 [-3.8; 11.67] | 6.11 [0.57; 18.4] |
| SFA, E% → P-Alb | -2.58 [-9.85; 4.58] | -4.07 [-11.26; 3.42] | -1.17 [-7.44; 4.97] | 5.14 [0.92; 12.4] |
| MUFA, E% → P-Alb | 1.81 [-6.77; 13.45] | -0.65 [-9.54; 8.54] | 3.92 [-3.14; 13.42] | 5.24 [1.04; 12.71] |
| Carbohydrates, E% → P-Alb | 1.29 [-14.77; 15.35] | 0.16 [-16.99; 15.88] | 3.71 [-11.99; 18.43] | 5.57 [0.76; 12.7] |
| Hydroxycholecalciferol → P-Alb | 1.71 [-6.25; 10.13] | 0.54 [-6.56; 8.47] | 3.47 [-6.7; 13.94] | 5.13 [0.43; 14.33] |
| Sodium → P-Alb | 1.22 [-8.68; 15.01] | -3.28 [-13.36; 7.35] | 2.8 [-5.8; 15.93] | 6.58 [1.71; 14.24] |
| Phosphorous → P-Alb | -0.19 [-7.16; 6.56] | -3.09 [-12.68; 4.94] | 1.73 [-4.98; 8.63] | 4.25 [0.7; 13.41] |
| Energy, kcal/kg → P-Alb | 1.60 [-11.49; 15.39] | -2.93 [-14.33; 8.17] | 2.15 [-12.02; 18.26] | 5.73 [0.38; 15.28] |
| Protein, g/kg → P-Alb | 0.33 [-12.64; 13.11] | -2.72 [-17.16; 10.61] | 1.62 [-12.93; 15.03] | 4.21 [0.54; 10.13] |
| Protein, E% → P-Alb | 0.64 [-7.3; 7.41] | 0.88 [-7.16; 8.59] | 2.67 [-4.8; 10.24] | 3.12 [0.09; 10.49] |
| Water → P-Alb | -2.04 [-6.16; 2.97] | -2.6 [-8.07; 2.01] | 0.11 [-3.6; 3.77] | 8.12 [0.78; 18.33] |
| Calcium → P-Alb | -0.08 [-10.35; 6.03] | -0.32 [-5.36; 4.17] | 2.6 [-3.21; 11.67] | 6.67 [0.5; 19.09] |
| PUFA, E% → P-Alb | -0.93 [-6.18; 5.86] | -2.42 [-6.01; 1.27] | -0.89 [-5.79; 4.35] | 4.89 [0.65; 11.59] |
| Vitamin D → P-Alb | 1.60 [-1.28; 4.01] | 0.11 [-3.45; 3.32] | 2.39 [-0.82; 6.41] | 4.12 [0.61; 10.44] |
| Fiber → P-Alb | 1.19 [-5.02; 7.55] | 0.7 [-3.35; 4.61] | 2.02 [-2.56; 7.51] | 4.06 [0.64; 12.82] |

The table presents the 20 effects of nutrients ($j$) on the considered concentrations ($m = 1, \ldots, 3$) which are strongest among the studied patients. Here, general effects ($\hat{\beta}_{jm}$) represent average responses across the patients studied ($k = 1, \ldots, 37$). Personal effects combine these general effects with variations from treatments ($\hat{g}_{ljm}, l = 1, \ldots, 3$) and individual variations within those treatments ($\hat{\beta}_{jm} + \hat{g}_{ljm} + \hat{b}_{kjm}$). Additionally, the table displays the estimated effect variation between dialysis treatments ($\hat{\sigma}_g$) and provides 90% credible intervals for all the parameters. All effects are estimated on an additive scale.

conditioning was done similarly to that of Fig 1B in which phosphorous and potassium were separated as the queried variables $Q_1$ and $Q_2$, which left the other nutrients, $X_j$, $j = 1, \ldots, 20$, unmodified. These queried variables were defined with the following proposal distributions

$$Q_1 \sim \text{Uniform}(0, 5800),$$
$$Q_2 \sim \text{Uniform}(0, 2550)$$

(9)

where the upper limits of the proposed intakes reflect maximum intakes in the general population. These limits were borrowed from the FinRavinto study [23] that studies the intake and nutrition of the Finnish adult population, and in which these limits are the maximum intake for 95% of studied subjects ($n = 565$). For the personalized concentration limits, given with the algorithm parameters $Y_{km}^l$ and $Y_{km}^u$, we used the normal ranges reported by laboratory staff and KDOQI guidelines [14]. The plasma albumin upper limit was personalized by the patient's age, as laboratory staff suggested. The exact ranges that were used are reported in Table 3. We executed the recommendation sampling with long chains of $S = 10, 000$ sample draws for each patient. Besides the means of nutrient level and nutrient effect distributions, the recommendation was executed also with their 5% and 95% quantile values for sensitivity analysis [24] of recommendations. The confidence level was also varied with the parameter options $c = 90\%$ and $c = 80\%$.

The algorithm resulted in a recommendation with $P_m^{max} > 90\%$ confidence for 22 of 37 patients, and 17 of those patients could exceed the general recommendations of potassium or phosphorous intake. For the rest of the patients, sufficiently confident recommendations could not be reached. When the 5% and 95% tail quantile values ($l_x$ and $l_\beta$) were used as estimations of current nutrient levels and the nutrient effect, only a few patients received confident recommendations, indicating that distributions for these variables were overly wide and would need further observations to achieve confidence for clinical use. Fig 4 illustrates the estimated recommendations for each patient; the first two columns display the recommended intake, and the following three columns offer the predicted concentration levels based on these recommendations. The reported intake recommendations were taken from the 5% and 95% quantile values in the estimated two-dimensional recommendation distributions. This reporting method allowed us to compare otherwise uneven recommendation distributions. It demonstrates that the estimated recommendations range from 1 to 5531 mg/d for potassium and 4 to 2527 mg/d for phosphorous with varying personal configurations. These personalized recommendations are considerably different than the general recommendations of 2500 mg/d for potassium and 1000 mg/d for phosphorous [8].

### Detailed analysis of personalized recommendations

From all the personalized recommendations in Fig 4, the estimated two-dimensional distribution for Patient 36 is expanded in Fig 3 for a closer analysis. The black dot on the intake recommendation plot indicates that the patient's current intake of potassium is approximately 2260 mg/d and phosphorous is approximately 860 mg/d. This intake configuration produced the patient's current plasma concentrations which are indicated by the dashed line at the concentration plots: 3.8 mmol/l plasma potassium, 1.4 mmol/l plasma phosphate, and 40.5 plasma albumin. To determine the personal ranges of intake recommendation, the patient's two-dimensional intake recommendation distribution offered simulated intake proposals that were predicted to take all three concentrations to normal ranges; the darkening colors indicate the increasing confidence of the estimation. The rectangle over this distribution defines the area where both intakes produce over 90% confidence in achieving the targeted concentrations.

This method for defining regions over uneven recommendation distributions was chosen for systematic reporting and comparison between patients. The rectangle in Fig 3 defines the same recommendation of 759–2702 mg/d for potassium and 693–1473 mg/d for phosphorous that was provided to Patient 36 in Fig 4. The rectangle's limits have matching colors with the predictive distributions of the concentrations. These four distributions are at the limits of concentration normal ranges and provide constraints for the recommendation.

The gray bars in Fig 4 indicate the counterfactual expected values of concentrations $\hat{\mu}_m^{q0}$ where the effects of potassium and phosphorous are totally omitted. This shows any available room in the concentrations to absorb the effects of phosphorous and potassium. For 7 of 22 patients who received confident recommendations, the initial concentrations were already over the upper limits, but the linear combinations of proposed intakes and their effects decreased the final concentrations within the normal range. For the rest of the patients, this initial level concentration was already within the normal range or lower than its lower limit. Especially for plasma albumin, this lower limit was difficult to reach for one-third of the studied patients.

## Evaluation of the personal models

Due to the limited number of personal observations, our primary purpose was to model the reactions of the patients in the studied data, but not to make generalizations about the reactions found. We developed several model candidates to explore the mechanisms in data. The models were evaluated with both visual posterior predictive checks [25, 26] and normalized root mean square errors (NRMSE) that were calculated over all concentration predictions $\hat{Y}_{kmi}$ for $K$ patients and $M$ concentrations with

$$\text{NRMSE} = \frac{\sum_{k=1}^{K} \text{NRMSE}_k}{K}, \ \text{NRMSE}_k = \frac{\sum_{m=1}^{M} NRMSE_{km}}{M}, \ \text{NRMSE}_{\text{km}} = \frac{\sqrt{\frac{1}{n}\Sigma_{i=1}^{n}\left(\hat{Y}_{kmi} - Y_{kmi}\right)^2}}{\overline{Y}_{km}} \ (10)$$

The posterior predictive check in Fig 5 compares the distribution of the measured concentrations with the samples drawn from the multivariate model. This check verified that the model is unbiased as the draws from the concentration model are centered with the measured concentrations. The predictions for phosphate and potassium concentrations have very little variance, but for albumin concentration, the prediction variance is higher. Additionally, we ran simulations with the personal graphical models by drawing samples from current diets and predicting personal concentrations. As no conditioning is done on the diets, the concentrations are expected to settle on their measured values. The differences between the simulated concentrations and the measured concentrations are estimated with the normalized root mean square errors (NRMSE) in Table 6. Here we considered an alternative effect model (mv3_cross_single_level) that omits the dialysis treatment as a separate layer in the hierarchical model but this increased the normalized error from 0.003 to 0.078. We also considered the concentrations as separate univariate models (separate_pk_fppi), but it increased the overall normalized error to 0.081 indicating the effect of multivariate modeling. All of these models were estimated with a Bayesian modeling framework Stan [21] with four chains and 3000 sample draws from each chain with a 1000 sample warm-up period. Model diagnostics confirmed that there were no divergent transitions during the sampling. The sampling produced an average effective sample size (ESS) of 580 over all model parameters. The chains are also considered mixed as R-hat convergence diagnostic is 1.01 on average. It is recommended that the effective sample size should be well over 100 and R-hat less than 1.05 [27] for parameters to be trusted.

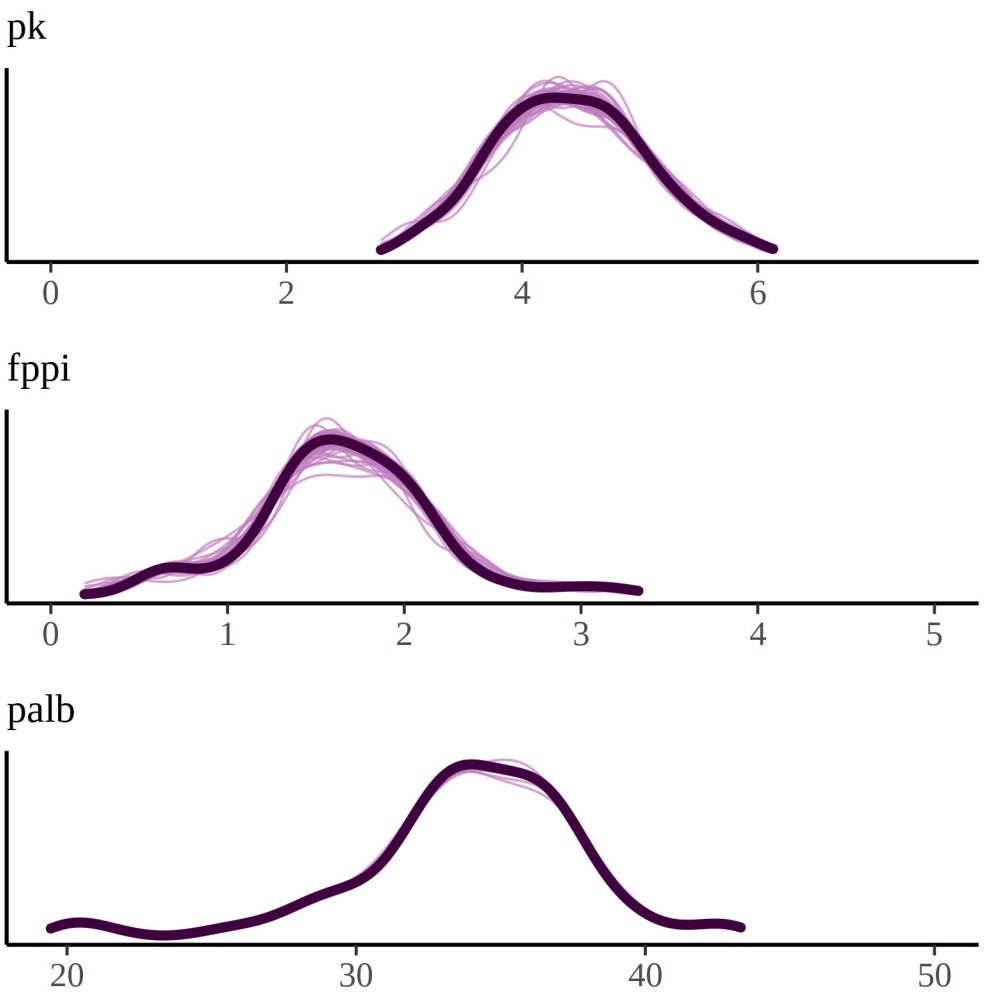

**Fig 5. Posterior predictive check (PPC) of the final nutritional effect model.** Posterior Predictive Check (PPC) for the final model version: Black lines represent observed concentration levels, while purple lines overlay the predicted concentration levels when all predictor nutrients match observed values. For an unbiased model, the observed and predicted concentrations are expected to be aligned. The figure is plotted with bayesplot package for R language (v 1.10, http://mc-stan.org/bayesplot/).

For external validation, we executed 10-fold cross-validation for the final version (mv3_cross_two_levels) of the model. In this process, we split the dialysis patient data into ten folds with three or four of all 37 patients removed from each fold. Personal reactions were predicted for these removed patients while the rest of the data were used for the model estimation.

**Table 6. Comparison of model structures for personal nutrient effects.**

| Reaction model | P-K | fP-Pi | P-Alb | average NRMSE |
|---|---|---|---|---|
| mv3_cross_two_levels | 0.003 | 0.006 | 0.002 | 0.003 |
| mv3_cross_two_levels_cv | 0.170 | 0.231 | 0.063 | 0.170 |
| mv3_cross_single_level | 0.078 | 0.233 | 0.103 | 0.078 |
| separate_pk_fppi | 0.081 | 0.175 | - | 0.081 |

The hierarchical model with two levels, personal and dialysis treatment levels, provided the best fit against the studied data as measured with normalized root mean square error (NRMSE). Together with the multivariate modeling of concentrations, it improved the fit in comparison to separately modeled concentrations.

The prediction was made with Eq (1) by applying the unseen concentration measurements $Y_{kmi}$ and corresponding nutrition data $X_{kjm}$ with the assumption that common effects ($\hat{\beta}_{jm}$) and dialysis treatments effects ($\hat{g}_{ljm}$) remained as previously modeled. Only the personal variations from the dialysis treatment averages were predicted to random variables $\hat{\mathbf{b}}_k = \hat{\mathbf{T}}_b\hat{\mathbf{L}}_b\mathbf{z}$ where the standard normal distribution $\mathbf{z} \sim \text{Normal}(\mathbf{0}, \mathbf{I})$ is transformed with the Cholesky decomposition $\hat{\mathbf{L}}_b$ from Eq (5) holding the learned correlations between nutritional effects. The related correlation matrix $\mathbf{C}_b = \mathbf{L}_b\mathbf{L}_b'$ is illustrated in S3 Fig with the highest correlations highlighted in S2 Table. Multiplying the distribution with $\hat{\mathbf{T}}_b$ adds the estimated standard deviations of personal effects to it. As a result, the prediction seeks to find the most probable combination of effects $\hat{\mathbf{b}}_k$ from distribution $\hat{\mathbf{T}}_b\hat{\mathbf{L}}_b\mathbf{z}$ given the observations from new patient $k$. We evaluated also the predicted concentrations with NRMSE and it increased to 0.170 from 0.003 in comparison with the in-sample model. In addition to the smaller amount of data, the increased error can also be explained with unseen patients that react differently than was estimated in the Choslesky matrix $\hat{\mathbf{L}}_b$ of that fold. S5 Fig illustrates the cross-validation folds of patients and compares the modeled and the predicted NRMSE values.

Finally, we conducted a sensitivity analysis [24] of recommendations by creating personal graphs and recommendations also from the cross-validation predictions. The predicted recommendations are shown in S4 Fig. In comparison to the in-sample recommendations of Fig 4, there are five patients whose predicted reactions did not produce recommendations confident enough, and three patients who could have recommendations with the predicted reactions but not with the in-sample model. All of these changes occurred in recommendations that are very close to the target limits already. The aim of the recommendation sensitivity analysis is to ensure that the recommendations are robust and react consistently.

## Discussion

The objective of this study was to develop a statistical method for deriving personalized intake recommendations that ensure plasma concentrations remain within or approach the normal ranges. We applied this method to infer recommendations for patients with end-stage renal disease (ESRD), a population at higher risk of malnutrition [1]. We chose to estimate the personalized levels of phosphorous and potassium as their greater intake would allow for a richer diet considering their strict general recommendations. Our decision aligns with the 2020 update of the KDOQI nutritional guidelines, which advocate for personalized adjustments to maintain normal ranges of plasma phosphate and potassium [7]. By applying our method, we provided a statistically justified inference of personalized dietary intake likely to result in normal plasma concentrations. Consequently, we observed considerable variations in the recommended dietary intake of phosphate and potassium for each patient.

### Limitations and reliability of the study

The data collected for this study had certain limitations, including a small sample size of patients ($n$ = 37) and a limited number of observations per patient. Each patient underwent only two dietary interviews and corresponding laboratory analyses of concentrations. The scarcity of data can be attributed to the fragility of renal patients undergoing dialysis, which poses challenges in conducting extensive studies involving this population. The main motivation of this study was to address the malnutrition-induced weakness experienced by these patients, but the subject is complicated to investigate, and thus literature on the nutritional reactions of renal disease patients is also limited. Given the small amount of data, it is not

recommended to generalize the findings of this study beyond the studied patients without further research.

However, we showed in Table 6 that including the patients' grouping in different types of dialysis treatments improved the personal models' fit considerably, and as a result, the multivariate model repeats the observed concentrations without bias in Fig 5. This makes the expected values of nutritional reactions solid to conclude that there exist personal differences within this patient population, and there is a need for personal inference of nutrition intake. Another source of uncertainty in the inference is the estimation of patients' current intake levels. In this partial recommendation, only levels of phosphate and potassium were conditioned, while the rest of the diet remained unmodified. For accurate recommendations, the contribution of the unmodified diet should be accurately estimated, and Algorithm (1) brings these contributions out with parameters $\hat{\mu}_m^{q0}$ to support the reasoning.

A major modeling decision was to choose whether the relations between the concentrations should be estimated, or if they should be considered independently. The latter provides a sparse and computationally efficient structure in a Bayesian network [28], but we have demonstrated that with this small amount of data, also multivariate computation can be accomplished. We conclude that if there exist cross-model correlations, then a multi-target recommendation requires a multivariate model of the reactions. In this analysis, sparse cross-model correlations are proven to exist, and including them enhanced the predictions as was seen in the decrease of normalized root mean square error.

## Application of the inference method

Our goal was to develop a recommendation method that enables practitioners to transparently follow the inference process. By estimating the nutrient components contributing to plasma concentrations, our method allows for inference regarding the hypothetical scenario where the intake of phosphorus and potassium is completely omitted. These concentration baseline parameters, denoted as $\hat{\mu}_m^{q0}$, serve as fixed starting points in the recommendation algorithm. Based on these baselines, the recommendation method offers three options for proceeding. In the first two options, the estimated baselines are either above or below the normal ranges of concentrations. If simulating intakes of phosphorus and potassium alone fails to reach the normal ranges, the recommendation is deemed unsuccessful. In the third option, if the baselines already fall within the normal range, it is still necessary to accumulate a sufficient probability mass from the predictive distributions of concentrations within the normal ranges to provide a confident recommendation. As the model gains more information about individual reactions, the predictive distributions are expected to become more tightly concentrated, with their expected values approaching the limits of the normal ranges. Consequently, the method can generate confident recommendations for more permissive diets. This concept is illustrated in Fig 3, where the predictive distributions of plasma potassium push against the boundaries of the normal ranges, thus constraining the corresponding intake recommendation. The breadth of these distributions signifies the uncertainty stemming from personalized estimations of diet and its effects. For reporting purposes, the recommendations presented in this study utilize the expected values of the predicted concentration distributions.

In addition to facilitating comparisons between patients, our method also supports personalized nutritional guidance, as exemplified in Fig 3. This figure depicts the current phosphorus and potassium intakes of Patient 36, along with the corresponding concentration levels. Despite all considered plasma concentrations falling within the normal ranges with the given intake levels, it is crucial to determine the boundaries of intake necessary to maintain concentrations within those ranges. Based on the personal model developed for Patient 36, Algorithm

(1) generates a recommendation for potassium intake ranging between 759 and 2702 mg/d, and for phosphorus intake ranging between 693 and 1473 mg/d. Notably, the expected concentration values for a simulated diet without potassium and phosphorus intake already fall within the normal ranges, along with substantial portions of the concentration predictive distributions. By comparing the corresponding colors of the intake recommendation limits and the predictive distributions of concentrations in Fig 3, we can deduce certain relationships. The concentration of plasma potassium (P-K) increases as potassium intake increases, while it also becomes evident that plasma potassium concentration increases with decreasing phosphorus intake. On the other hand, phosphorus intake has minimal impact on plasma phosphate concentration. However, a decrease in phosphorus intake results in an increase in plasma albumin concentration (P-Alb). Together, these reactions define the lower limit of the phosphorus intake recommendation and the upper limit of the potassium intake recommendation. As phosphorus intake increases and potassium intake decreases, the lower limits of the normal ranges for potassium and albumin concentrations are first met. For many patients, the recommendation is constrained by the upper limit of fasting plasma phosphate and the lower limit of plasma albumin. In clinical practice, making choices among different diet options that yield computationally identical concentrations requires nutritional expertise and may vary between patients, depending on individual factors.

## Considerations on the personal effects of nutrients

In the partial recommendation presented in this study, only the levels of phosphate and potassium were specifically adjusted, while the remainder of the diet was left unmodified. The factors taken into account for personalized recommendations were the estimated nutritional reactions and the composition of the current diet. The literature on the nutritional reactions of renal disease patients is scarce, but our hierarchical reaction model indicated that incorporating the type of dialysis treatment as a nested level of grouping improved the accuracy of the model. While there are differences in the average reactions among patients receiving different dialysis treatments, individual variations in reactions were also substantial for many factors. Notably, the average effect of water intake on plasma albumin showed the greatest variation among the different dialysis treatments. Patients undergoing hospital hemodialysis exhibited weaker reactions compared to those undergoing home or peritoneal dialysis. This difference is likely attributed to the less frequent occurrence of hospital hemodialysis and the treatment guidance in hospital settings that restricts fluid and sodium intake [7][14]. Our results also revealed weaker sodium reactions in hospital dialysis.

It is worth noting that the nutrient effects on plasma albumin concentration exhibited the greatest variability among patients. However, it is important to interpret this result cautiously, as the posterior predictive check in Fig 5 indicated a high variance in the model predictions for plasma albumin concentration, despite the predictions being unbiased. We attribute this high variance to the fact that plasma albumin concentration cannot be fully predicted based on the nutrient intake. In fact, it is reported that plasma albumin and prealbumin concentrations should not be relied upon as exclusive nutritional markers [29]. These concentrations may decrease in the presence of inflammation, regardless of the underlying nutritional status. Upon treating the malnutrition, the inflammation may subside, leading to an increase in albumin concentration. Polyunsaturated fatty acids (PUFA) are known to have inflammatory properties [30] and can aid in the recovery process. However, the specific effect of PUFA on albumin concentration has been shown to vary among patients. Thus, the composition of albumin concentration is more intricate compared to the other considered concentrations. Lastly, the intakes of both phosphorus and potassium also exhibit individually varying effects on all

the concentrations, as presented in Table 4. Addressing this variation is our primary contribution and is reflected in the resulting personalized recommendations.

## Future work

Currently, Algorithm (1) requires that all concentration targets must be fully reached before providing a recommendation. However, it is possible to allow patients to partially reach the targets, which would necessitate defining an order of importance among the concentrations. In such a scenario, achieving the most important concentration would be required, while the less important concentrations would need to be reached to the greatest extent possible. In our future work, we intend to personalize the intake recommendations for all nutrients or as many as practically feasible. We have observed that modifications in phosphorus and potassium levels alone were not sufficient for all patients. However, this poses computationally a more challenging problem, as there will be numerous dimensions to consider in the recommendation distribution beyond these two nutrients. Currently, the recommendation algorithm provides parameters that can be used to track the inference process. However, collecting and interpreting the relevant parameters may be a tedious task. Therefore, in our future work, we aim to enhance the interpretability of the recommendations by incorporating an algorithmic explanation alongside them. This will help users better understand the reasoning behind the personalized recommendations and facilitate their implementation in clinical practice.

## Conclusion

In conclusion, this work shows that end-stage renal disease (ESRD) patients are unique in many ways. These patients exhibit varying responses to nutrients, follow diverse dietary patterns, and even possess distinct targeted normal ranges for plasma concentrations. The task of providing well-grounded intake recommendations becomes challenging without the aid of computational tools. Fortunately, probabilistic graphical models have proven to be a viable approach for personalized modeling. These models offer flexibility in diet inference, while the Bayesian framework empowers practitioners to effectively manage uncertainty in estimations. Moreover, the inclusion of a hierarchical model parameterization enables the graphical models to provide recommendations that can be tailored to more granular levels, including population, treatment, and individual preferences. Our contributions can be summarized in the following:

- This study provides evidence that there exist considerable and quantifiable differences in how renal patients' plasma concentrations react to the same nutrition.

- As also the patients' diets differ, personally different phosphorous and potassium intakes are needed for maintaining normal plasma phosphate and potassium concentrations.

- Bayesian inference provides a systematic method for personal phosphorous and potassium recommendations and could support clinical nutritionists in providing personalized guidance.

## Supporting information

**S1 Fig. Intake levels of potassium and phosphorous from the collected food records and their observed effects on these concentrations.** The recommended normal ranges are denoted with white areas.
(TIF)

**S2 Fig. Correlation plot of matrix $C_g$ that includes the estimated within-model and cross-model correlations in level of dialysis treatment.** The figure is plotted with ggcorrplot

package for R language (v 0.1.4, https://cran.r-project.org/web/packages/ggcorrplot).
(TIF)

**S3 Fig. Correlation plot of matrix C$_b$ that includes the estimated within-model and cross-model correlations of personal effects.** The figure is plotted with ggcorrplot package for R language (v 0.1.4, https://cran.r-project.org/web/packages/ggcorrplot).
(TIF)

**S4 Fig. Personal recommendations of potassium and phosphorous intake ($\hat{Q}^{min} - \hat{Q}^{max}$) based on cross-validation predictions of nutrition effects, $\hat{b}_k$. Other parameters of the figure are similar to in-sample prediction in Fig 3.** The figure is plotted with ggplot2 package for R language (v 3.4.1, https://ggplot2.tidyverse.org).
(TIF)

**S5 Fig. Figure shows normalized root mean square error (NRMSE) of the two-level model for each patient.** Blue points indicate the in-sample error and red points indicate the error from cross-validation prediction. The lines between the points highlight the increased error between in-sample modeling and prediction. Alternating colors of the lines denote the folds of the cross-validation. The figure is plotted with ggplot2 package for R language (v 3.4.1, https://ggplot2.tidyverse.org).
(TIF)

**S1 Table. Table shows 40 highest positive or negative correlations between potassium and phosphorous treatment effects with other treatment effects.** This structure of correlations is used in estimating the personal effects based on personal intake and matching concentrations.
(PDF)

**S2 Table. Table shows 40 highest positive or negative correlations between personal effects of potassium and phosphorous with other personal effects.** This structure of correlations is used in estimating the personal effects based on personal intake and matching concentrations.
(PDF)

**S3 Table. All the estimated nutrition effects in general, dialysis treatment and personal levels.** Nutrition effect magnitudes from nutrients and other modeled features ($j = 1, \ldots, 22$) to blood concentrations ($i = 1, \ldots, 3$) for analyzed patients ($p = 1, \ldots, 37$) in all three additive levels of the model. General effects ($\hat{\beta}_{ij}$) are shown to vary between patients in home hemodialysis, hospital hemodialysis, and peritoneal dialysis. The first column of each dialysis type (avg) shows the typical effect of the treatment ($\hat{\beta}_{ij} + \hat{g}_{ijk}, k = 1, \ldots, 3$) that can further vary personally. Minimum and maximum of these personal effects are shown within each treatment ($\hat{\beta}_{ij} + \hat{g}_{ijk} + \hat{b}_{ijp}$). The table is sorted in decreasing order of between-treatment variation ($\hat{\sigma}_g$) and all the estimates include their 90%-credible intervals.
(PDF)

**S4 Table. Summary of notation for Method section.**
(PDF)

## Acknowledgments

Part of the work was done while VH was affiliated to the Department of Electrical and Computer Engineering, National University of Singapore.

## Author Contributions

**Formal analysis:** Jari Turkia.

**Methodology:** Jari Turkia.

**Software:** Jari Turkia.

**Supervision:** Ursula Schwab, Ville Hautamäki.

**Validation:** Ursula Schwab, Ville Hautamäki.

**Visualization:** Jari Turkia.

**Writing – original draft:** Jari Turkia.

**Writing – review & editing:** Ursula Schwab, Ville Hautamäki.

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
