## [Decision Letter · Decision Letter 0]

27 Sep 2023

PONE-D-23-20128Inferring personal intake recommendations of phosphorous and potassium for end-stage renal failure patients by simulating with Bayesian hierarchical multivariate modelPLOS ONE

Dear Dr. Turkia,

Thank you for submitting your manuscript to PLOS ONE. After careful consideration, we feel that it has merit but does not fully meet PLOS ONE’s publication criteria as it currently stands. Therefore, we invite you to submit a revised version of the manuscript that addresses the points raised during the review process.

We look forward to receiving your revised manuscript.

Kind regards,

Willi Jahnen-Dechent

Academic Editor

PLOS ONE

Journal Requirements:

**Additional Editor Comments:**

Your paper has been seen by a member of the Editorial Board and one outside reviewer. Both agree that the paper has merit, but in its current form requires too much knowledge of statistics and bioinformatics to be of use for patients and doctors who will mostly benefit from personalized nutritional recommendations. Please respond to all “review comments to the author” and improve writing and figures as suggested.

Reviewers' comments:

Reviewer's Responses to Questions

**Comments to the Author**

1. Is the manuscript technically sound, and do the data support the conclusions?

Reviewer #1: Yes

2. Has the statistical analysis been performed appropriately and rigorously? 

Reviewer #1: I Don't Know

3. Have the authors made all data underlying the findings in their manuscript fully available?

Reviewer #1: Yes

4. Is the manuscript presented in an intelligible fashion and written in standard English?

Reviewer #1: No

5. Review Comments to the Author

Reviewer #1: In terms of personalized medicine an approach to adjust dietary recommendations is highly interesting, especially for the management of potassium, phosphate and protein in dialysis patients, which is still a big issue in nephrology. However, the provided manuscript in its current version might be difficult to understand for the medical community and requires modifications. This accounts especially for the graphical models with additional explanations and more guidance is needed how to apply the models on the personalized diet recommendations. Most figure designs are quite challenging and should be simplified and structured more clearly to address the readers.

How were the variables selected that were implemented in the Bayesian model (Serum concentrations and nutrient predictors)? This is especially of interest as, also stated by the authors, there is only limited direct associations between phosphate, protein and potassium.

How was the time interval between blood sampling and dialysis for both blood collections? Dialysis treatment will adjust for serum potassium and in parts for serum phosphate. A small scheme depicting the workflow, and including the interview parts would be helpful. For phosphorous it is mentioned that patients were fasted for potassium and for albumin not (which would be preferred)?- Does that mean there were different sampling timepoints for each visit? And if so, what was eaten in between?

Not all parameters/ abbreviations seem to be explained in the description of the personalized graphical models and the unrelated regressions. These sections are very difficult to read and should be rephrased. A table explaining all variables (e.g. as supplementary table) could help.

I appreciate the idea showing a personalized recommendation for one single patient (figure 2). However the given association with albumin is difficult to understand. The 3 small graphs are quite confusing- if intakes and serum levels are depicted in one graph, this needs to be stated more clearly. Should dietary protein be considered here? The benefit of the model for the personalized recommendation should be worked out more clearly (e.g. if the patient increases intake of x, and increase in y becomes more likely..) and the graph requires a more clear structure and explanations of the color code.

Table 5 seems very relevant and could help to improve therapeutical options. But why was the intrapersonal effect versus the inter-patient variation shown and in supplementary table 3 it was stratified by type of dialysis? The general effect should be implemented into the main figure.

What do the green marks in figure 3 indicate? Is this the recommended range per parameter? Why are ranges (first columns) and recommendations (column 3-5) mixed in one graph? Again, quite confusing to a reader. Why not showing recommended daily nutrient intakes as well?

Figure 4: The descriptions in the posterior predictive check are too complicated- what is the observed data and what is the predicted one (black line? Purple line?)

6. PLOS authors have the option to publish the peer review history of their article (what does this mean?). If published, this will include your full peer review and any attached files.

Reviewer #1: No

---

## [Author Response · Author response to Decision Letter 0]

14 Nov 2023

Dear Editor,

Thank you for the review and valuable comments. We have acknowledged your concerns about the paper requiring too much statistical background information and have done our best to clarify the writing and figures for the paper to be most valuable to readers. We have addressed your concerns point-by-point as follows.

Journal Requirements:

Response > We have ensured that the manuscript follows these style requirements:

In this revision we have made following stylistic changes:

- Corresponding author is denoted as in the guide:

* Corresponding author

E-mail: jari.turkia@cgi.com (JT)

- File naming is according to the requirements.

Response > The minimal data set has been available in a Github repository so that it works with the supplementary source code, and we have now uploaded the minimal data set also in the Supporting Information files for easier access.

Response> We have added details about the IRB, Kuopio University Hospital Research Assistance Center ("KYS Tiedepalvelukeskus" in Finnish), and the waived ethics approval to the Dialysis patient data section. The section also mentions the written informed consent that was obtained.

Reviewer 1

In terms of personalized medicine an approach to adjust dietary recommendations is highly interesting, especially for the management of potassium, phosphate and protein in dialysis patients, which is still a big issue in nephrology. However, the provided manuscript in its current version might be difficult to understand for the medical community and requires modifications. 

This accounts especially for the graphical models with additional explanations and more guidance is needed how to apply the models on the personalized diet recommendations. 

Most figure designs are quite challenging and should be simplified and structured more clearly to address the readers.

1. How were the variables selected that were implemented in the Bayesian model (Serum concentrations and nutrient predictors)? This is especially of interest as, also stated by the authors, there is only limited direct associations between phosphate, protein and potassium.

Response> The aim of the analysis was to explore the possibility of less restricted phosphorous and potassium intake while making sure that their concentrations stayed in their normal ranges. All the other nutrient predictors in the model were selected to reflect patients’ current diet. We assumed all the energy nutrients, vitamins, minerals, and fluids to be relevant in the determining the concentrations. Possible similar effects of nutrients (collinearity) were mitigated with QR-decomposition in the model. This is now clarified in the data section: 

“Laboratory tests for renal patients included several measurements, from which concentrations of plasma potassium (P-K), fasting plasma phosphorous (fP-Pi), and plasma albumin (P-Alb) were selected as targets of this analysis for exploring the possibility of less restricted phosphorous and potassium intake. The selected predictors were assumed to reflect the composition of these concentrations; all the energy nutrients, vitamin D, minerals, and fluids. Also, the selected medications were known to directly affect the concentrations. Patients fasted before the laboratory tests although there were analyses that did not require fasting. The schema for data collection is outlined in Fig. 1.”

2. How was the time interval between blood sampling and dialysis for both blood collections? Dialysis treatment will adjust for serum potassium and in parts for serum phosphate. A small scheme depicting the workflow, and including the interview parts would be helpful.

Response> The laboratory tests always occurred before dialysis treatment. This is now clarified in the text. We appreciate the suggested scheme for data collection workflow, and it is now added in the revision.

3. For phosphorous it is mentioned that patients were fasted for potassium and for albumin not (which would be preferred)?- Does that mean there were different sampling timepoints for each visit? And if so, what was eaten in between?

Response> All the analyses were taken at the same timepoints, and patients fasted before the laboratory tests, but for example, P-Alb does not require fasting, so it is not marked in the notation. Fasting is also clarified now in the text and noted in the workflow scheme.

4. Not all parameters/ abbreviations seem to be explained in the description of the personalized graphical models and the unrelated regressions. These sections are very difficult to read and should be rephrased. A table explaining all variables (e.g. as supplementary table) could help.

Response> We have elaborated the key concepts in this section more carefully and made sure that all the variables and notations are explained in the text. The text is also revised for better clarity. Supplementary Table S4 has also been added to summarize the variables and other notations of the method section.

5. I appreciate the idea showing a personalized recommendation for one single patient (figure 2). However the given association with albumin is difficult to understand. The 3 small graphs are quite confusing- if intakes and serum levels are depicted in one graph, this needs to be stated more clearly. Should dietary protein be considered here? The benefit of the model for the personalized recommendation should be worked out more clearly (e.g. if the patient increases intake of x, and increase in y becomes more likely..) and the graph requires a more clear structure and explanations of the color code.

Response> In Figures 2 and 3, one key aspect is to show the recommended nutrient ranges and the concentrations that the recommendations are predicted to produce. Keeping them both in the same Figures aims to show the function of our method. We clarified the figures in this revision and added explanations of the color coding that binds these subfigures.

Dietary protein would possibly affect the modeled concentrations, but in this study, we chose to consider only phosphorous and potassium intakes and the possibility of less restricted intake. As stated in Discussion, modifying these nutrients alone is not enough for every patient, and in our future work our aim is to generalize the present method to modify the whole diet. Biological interpretation of the model, in context of Fig. 2, is also worked out in the Discussion section. In this revision, we have emphasized in Fig 4 the patients who could be recommended a less restrictive diet. This is the main benefit of the model.

6. Table 5 seems very relevant and could help to improve therapeutical options. But why was the intrapersonal effect versus the inter-patient variation shown and in supplementary table 3 it was stratified by type of dialysis? The general effect should be implemented into the main figure.

Response> We have added the general effect to Table 5 and removed the intra-patient variation as becomes clear from the already presented minimum and maximum of personal effects. Treatment induced variation is still shown as due to different principles of action it affects the nutrient effects and is a relevant predictor.

7. What do the green marks in figure 3 indicate? Is this the recommended range per parameter? Why are ranges (first columns) and recommendations (column 3-5) mixed in one graph? Again, quite confusing to a reader. Why not showing recommended daily nutrient intakes as well?

Response> The green marks in the potassium and phosphorous columns indicate the personally recommended range of intake, and the numbers in those columns indicate the 90%-quantile values of the range. We agree that the Figure is busy and we have now removed some unnecessary information like confidence percentages of predictions. The intake columns already show the general recommendations of potassium and phosphorous intake (with solid black lines), and the green marks have been changed to blue and red color-coding to indicate if personal recommendation exceeds the general recommendation. This is the key takeaway from this Figure. 

It is essential to keep the intake recommendations and the corresponding concentration predictions in the same Figure as the concentration predictions induce the recommendations. Splitting these in separate Figures would make this connection hard to trace. We have added more legends to the Figure and rephrased the caption to make this connection clearer.

8. Figure 4: The descriptions in the posterior predictive check are too complicated- what is the observed data and what is the predicted one (black line? Purple line?)

Response> We have simplified the caption for the posterior predictive check: “Posterior Predictive Check (PPC) for the final model version: Black lines represent observed concentration levels, while purple lines overlay the predicted concentration levels when all predictor nutrients match observed values. For an unbiased model, the observed and predicted concentrations are expected to be aligned.”

Sincerely,

Mr. Jari Turkia, 

Corresponding Author

On behalf of all the authors

---

## [Editor Report · Decision Letter 1]

2 Jan 2024

Inferring personal intake recommendations of phosphorous and potassium for end-stage renal failure patients by simulating with Bayesian hierarchical multivariate model

PONE-D-23-20128R1

Dear Dr. Turkia,

We’re pleased to inform you that your manuscript has been judged scientifically suitable for publication and will be formally accepted for publication once it meets all outstanding technical requirements.

Kind regards,

Willi Jahnen-Dechent

Academic Editor

PLOS ONE

Additional Editor Comments (optional):

The points raised during review have all been addressed. The manuscript has been modified accordingly.
---

## [Editor Report · Acceptance letter]

26 Jan 2024

PONE-D-23-20128R1 

PLOS ONE

Dear Dr. Turkia, 

I'm pleased to inform you that your manuscript has been deemed suitable for publication in PLOS ONE. Congratulations! Your manuscript is now being handed over to our production team.

Kind regards, 

on behalf of

Professor Willi Jahnen-Dechent 

Academic Editor

PLOS ONE